# Improving the Efficacy of Mesenchymal Stem/Stromal-Based Therapy for Treatment of Inflammatory Bowel Diseases

**DOI:** 10.3390/biomedicines9111507

**Published:** 2021-10-20

**Authors:** Mercedes Lopez-Santalla, Marina Inmaculada Garin

**Affiliations:** 1Division of Hematopoietic Innovative Therapies, Centro de Investigaciones Energéticas, Medioambientales y Tecnológicas (CIEMAT) and Centro de Investigación Biomédica en Red de Enfermedades Raras (CIBER-ER), 28040 Madrid, Spain; 2Advanced Therapy Unit, Instituto de Investigación Sanitaria Fundación Jiménez Díaz (IIS-FJD/UAM), Building 70, Floor 0. Avda. Complutense, 40, 28040 Madrid, Spain

**Keywords:** inflammatory bowel disease, mesenchymal stem/stromal cells, improvement protocols

## Abstract

Inflammatory bowel diseases (IBD) consisting of persistent and relapsing inflammatory processes of the intestinal mucosa are caused by genetic, environmental, and commensal microbiota factors. Despite recent advances in clinical treatments aiming to decrease inflammation, nearly 30% of patients treated with biologicals experienced drawbacks including loss of response, while others can develop severe side effects. Hence, novel effective treatments are highly needed. Mesenchymal stem/stromal cell (MSCs) therapy is an innovative therapeutic alternative currently under investigation for IBD. MSCs have the inherent capacity of modulating inflammatory immune responses as well as regenerating damaged tissues and are therefore a prime candidate to use as cell therapy in patients with IBD. At present, MSC-based therapy has been shown preclinically to modulate intestinal inflammation, whilst the safety of MSC-based therapy has been demonstrated in clinical trials. However, the successful results in preclinical studies have not been replicated in clinical trials. In this review, we will summarize the protocols used in preclinical and clinical trials and the novel approaches currently under investigation which aim to increase the beneficial effects of MSC-based therapy for IBD.

## 1. Introduction

Inflammatory bowel disease (IBD) consists of a chronic inflammation disorder which involves the intestinal mucosa of digestive tract. IBD includes Crohn’s disease (CD) and ulcerative colitis (UC). CD affects the entire digestive tract, often in a non-contiguous manner, with transmural inflammation leading to complications such as fibrotic structures, fistulas, and abscesses. UC mainly affects the colon with superficial mucosal inflammation and extends proximally in a contiguous manner leading to ulcerations and severe bleeding [1]. IBD predominantly affects young individuals, with a prevalence in western countries up to 0.5% of the general population, with growing incidence [2]. The clinical manifestations of IBD include abdominal pain, diarrhoea, haematochezia, reduced appetite, vomiting, loss of weight, and fatigue, with alternating flares and periods of remission which seriously affect patients’ quality of life [2]. The risk of colon cancer in these patients due to chronic inflammation is very high; therefore, the incidence of mortality is even higher in these patients [1,3].

The aetiology of IBD is not clearly established, although several aspects such as genetic and environmental factors, together with the immune responses against the epithelium and the commensal flora have been described [1]. CD is considered to be a predominant type 1 T helper cell (Th1)- and Th17-mediated disease with an increased production of interleukin (IL)17, interferon (IFN) γ and tumour necrosis factor (TNF) α; while UC has been associated with a dysregulated Th2 response [2].

Current treatments for IBD aim to decrease inflammation to prevent recurrence and to prolong periods of remission. These treatments include untargeted therapies such as aminosalicylates (principally mesalazine), glucocorticoids (conventional and other forms such as budesonide or beclomethasone), antibiotics (typically ciprofloxacin and metronidazole), and immunosuppressants (mostly azathioprine/6-mercaptopurine or methotrexate), as well as targeted therapies such as biologic therapies, which principally include TNFα inhibitors (infliximab, adalimumab, certolizumab pegol, and golimumab) [4] and blockers of signal transduction cascades (JAK inhibitors). More recently, IL12 and IL23 antibodies that drive the differentiation and function of specific immune cells (ustekinumab) [5] and modulators of lymphocyte trafficking (anti–α4β7 integrin antibodies such as vedolizumab) have been approved for IBD patients [2]. Together with these biological treatments, the use of hematopoietic stem cell (HSCs) transplantation has also been explored in some severe forms of gastrointestinal diseases, including IBD [6].

Despite these advances, nearly 30% of patients do not respond to current treatments for IBD, and 50% of them suffer allergic reactions or become refractory over time [1,2]. Moreover, these treatments are expensive, can lead to reactivation of infections such as tuberculosis or hepatitis B, thus increasing the risk of some cancers [3]. Thus, the need for surgery remains high; the risk of surgery 10 years after diagnosis is still 16% in UC and even higher, up to 47%, in CD patients. Consequently, the development of new therapeutic treatments for IBD is urgently required. In this context, mesenchymal stem/stromal cell (MSCs) therapy may be an innovative therapeutic alternative for IBD due to its capacity for modulating inflammatory immune responses and tissue regeneration [7]. At present, a significant number of preclinical studies in IBD have shown that systemic administration of MSCs can reduce intestinal inflammation without adverse effects [8,9] although these successful results in animal models of IBD have not been replicated in phase I/II clinical trials [10]. By contrast, the use of MSC-based therapy to treat complex perianal fistulas in CD has led to very successful results [11] allowing the approval of Alofisel^®^ (darvadstrocel/Cx601, Takeda) by the European Medical Agency as the first cell-based treatment composed of allogeneic adipose-derived MSCs (allo-ASCs) for perianal fistulizing Crohn’s disease (PFCD) [12,13].

This review provides a detailed overview of protocols and technical approaches currently under development which aim to increase the beneficial effects of MSC-based therapy in preclinical studies of IBD.

## 2. Search Strategy and Selection Criteria

The large majority of published data up to June 2021 which used MSC therapy to treat IBD in animal models have been included. We have used the following terms: ’mesenchymal stromal cells´ or ´mesenchymal stem cells´ or ´mesenchymal´ and ´stromal´ and ´cells´ or ´mesenchymal´ and ´stem´ and ´cells´ and ´colitis´ or ´inflammatory bowel disease´ or ´inflammatory´ and ´bowel´ or ´intestinal inflammation´ or ´dss´ or ´tnbs´ or ´adoptive T-cell transfer´ or ´acetic acid´. We searched in the following electronic databases: PubMed, Web of Science, patent registry (https://patentscope.wipo.int, (accessed on 30 June 2021)), Clinicaltrials.gov, Clinicaltrialregister.eu, and Trialregister.nl. Our electronic search strategy excluded non-English articles, in vitro studies and human IBD sample studies.

## 3. Characteristics of Preclinical Studies of IBD with Systemic MSC Therapy

### 3.1. Animal Models

For more than three decades, numerous animal models of intestinal inflammation have provided valuable insights into the occurrence of IBD. No single animal model of intestinal inflammation mimics all aspects of human IBD, as the specific pathogenesis underlying IBD is highly complex [14]. The large majority of MSC-based therapy preclinical studies for IBD have been carried out in chemically induced colitis models using dextran sulphate sodium (DSS) (Appendix A) and 2,4,6-trinitrobenzenesulfonic acid (TNBS)-induced colitis (Appendix A) for the induction of intestinal inflammation in mice and rats with very successful outcomes. In these models, the onset of inflammation occurs soon after induction, and the procedures required are technically feasible. These models have allowed the identification of genes and immunological mechanisms involved in IBD susceptibility during the development of acute, recovery, and chronic phases of colitis. One cycle of 3% to 5% DSS administration for 4–10 days in drinking water results in acute injury with partial crypt depletion on the distal segment of colon with multiple skipped mucosal erosive lesions together with an inflammatory cell infiltration of mononuclear/polymorphonuclear cells. After the late recovery phase, macrophages and CD4^+^ T cells become prominent in areas of wound healing in the basal portion of the lamina propria. Furthermore, several DSS cycles mimic the chronicity of human IBD. Similar to DSS, rectal administration of TNBS with ethanol induces severe transmural colitis, which is mediated by Th17 and Th1 immune responses [14].

### 3.2. Tissue Sources, Routes of Administration, and MHC Context

Mesenchymal stem/stromal cells are defined by the International Society for Cellular Therapy as plastic adherent cells that express CD105, CD90, and CD73, but not pan-leukocyte (CD45), endothelial (CD31), or primitive hematopoietic (CD34), monocyte (CD14 or CD11b), or B cell (CD79a or CD19) markers, as well as the human leukocyte antigen class II (HLA-DR) surface antigen [15]. Bone marrow, adipose tissue, and umbilical cord are the most common tissue sources used to isolate and expand MSCs. Intravenous (IV) and the intraperitoneal (IP) are the most frequently used routes of administration. Some studies have also used local administration. No clear differences in the beneficial effects of MSC-based therapy regarding the different tissue sources, major histocompatibility complex (MHC) contexts, and routes of administration have been reported (Appendix A). In the completed and ongoing clinical trials, MSCs from bone marrow (BM) is the most common tissue source used, followed by umbilical cord (UC) and adipose tissues (AD)-MSCs (Appendix A) [10,16]. In contrast to preclinical studies, the use of allogeneic MSCs is the chosen MHC context in the registered IBD clinical trials thanks to the documented low immunogenicity of the MSCs that allows cell bank establishment. This is in contrast to the autologous use of MSCs that requires in vitro expansion for several weeks prior to administration, thus adding time to patient care. No evidence regarding superior therapeutic potential or toxicity profile of one MHC context over the other exists [10]. IV administration is the most common route of administration currently used (Appendix A). Interestingly, in one of the completed clinical trials (NCT01221428; Appendix A) two doses of MSCs were infused into patients with moderate to severe UC one week apart. The first dose of MSCs was conducted by IV infusion, while the second dose of MSCs was administered via the superior mesenteric artery using interventional catheterization. According to published results, no adverse side effects were observed during the follow up period of the study (2 years) and beneficial effects of the MSC-based therapy were reported in those patients treated with MSCs with respect to the control group [17]. Ongoing clinical trials using MSC local administration is currently being tested (see Appendix A).

Although not yet in the clinic, several investigators have proposed alternative tissue sources for MSC isolation and expansion, such as amniotic and menstrual fluids and from tonsil, gingiva, and endometrial tissues. All of these tissues have similar characteristics to BM-MSCs, AD-MSCs, and UC-MSCs. Additionally, we and other investigators have conducted preclinical studies of IBD using novel routes of administration such as intranodal [18], subcutaneous [19,20], and colonic submucosa [21] for MSC infusion with very promising results. These could potentially be applicable for cell-based therapies in IBD (Appendix A and Table 7).

It is of interest that, although the biodistribution of MSCs is strongly dependent on the route of administration used, no clear correlation between the in vivo biodistribution of the MSCs and their capacity to modulate pathological immune responses has been established [18,22,23]. The general assumption is that the therapeutic action of MSCs is transient [24]. In contrast to this, we and others have reported sustained beneficial effects of MSC-based therapy both in preclinical [25,26,27] and in clinical trials [28,29,30].

### 3.3. Cell Dosage and Schedule of Infusion

As in other immune-mediated disorders, such as rheumatoid arthritis [31], several preclinical studies in IBD with MSC-based therapy have observed improved efficacy when the MSC infusion was performed during the early phases of the disease rather than at pre-onset or during chronic phases of inflammation induced by treatment with DSS [32,33,34] or TNBS [35,36]. Similar observations have also been made when MSC conditioned media [33] or even MSC-derived exosomes [35] were used (Appendix A, Table 5 and Table 6). In contrast, most of the completed IBD clinical trials with MSC-based therapy have been conducted in refractory CD and UC patients at advanced stages of the disease which may have hindered the beneficial effects of MSC-based therapy in the clinic (Appendix A). Interestingly, IBD patients during the early phases of the disease are now being recruited in currently ongoing clinical trials similarly to other immune-mediated disorders such as rheumatoid arthritis [37]. This change in the inclusion criteria may increase the efficacy of MSC-based therapy (Appendix A).

A great range of MSC dosages have been tested in preclinical IBD studies, from 0.01 × 10^6^ up to 30 × 10^6^ MSCs per animal in a single or multiple infusions (up to 5), although the most commonly used MSC dose has been in the range of 1 × 10^6^ to 10 × 10^6^ MSCs per mouse in a single dose (Appendix A). No adverse effects have been reported in any of the MSC doses tested. MSC dosage escalating to human use should be between 0.5 × 10^6–^1500 × 10^6^ MSCs/Kg based on mouse (≈20 g) and human (≈70 Kg) body weights. In completed IBD clinical trials, a range of 140 to 2400 × 10^6^ MSCs per patient in a single or multiple infusions (up to 12) have been used, higher than the MSC dosage used in the preclinical studies. In ongoing clinical trials, lower ranges of MSC dosages are being used (from 15–800 × 10^6^ by IV route of administration) together with the use of local administration of MSCs in the 50–600 × 10^6^ MSC range (Appendix A).

### 3.4. Mechanisms Involved in the Therapeutic Effects of MSC-Based Therapy

Different immune responses and mechanisms of action have been implicated in the immunomodulatory capacity of MSCs in preclinical studies of IBD. Among these, macrophage reprogramming, induction of both regulatory innate and adaptive immune cells, inhibition of inflammatory cells such as T helper 1 (Th1), Th17, and dendritic cells, secretion of molecules with anti-inflammatory effects such as indoleamine 2,3-dioxygenase (IDO) and prostaglandin E2 (PGE2), and induction of transforming growth factor beta (TGF-β1) and IL10 have been described (Appendix A). Additionally, in preclinical models of IBD, several groups have carried out experiments with MSCs deficient in different molecules aiming to further delineate additional molecular mechanisms involved in the beneficial effects of MSC-based therapy in experimental colitis such as FASL [38], tumour necrosis factor-inducible gene 6 protein (TSG-6) [20,39,40,41,42] or their extravesicles (EVs) [43], IL-1 β receptor [44], cystathionine β-synthase (Cbs) [45], Jagged-1 and TLR3 [46], autoimmune regulator (Aire) [47], insulin-like growth factor binding protein 7 (IGFBP7) [48], thrombospondin-1 (TSP-1, TGF-β activator) [49], matrix Gla-protein (MGP) [50], or CCL2 [51] (Appendix A). Other groups have used an array of inhibitors as a very useful tool to dissect the mechanisms involved in the therapeutic effects of MSCs. In this sense, Park HJ et al. used celecoxib, a known inhibitor of cyclooxygenase (COX-2) [52]. Alternatively, Kim HS et al. [53], who used a ribonucleic acid (RNA) interference to nucleotide-binding oligomerization domain-containing protein 2 (NOD), demonstrated the key role played by the NOD2 pathway in UC-MSCs to modulate experimental colitis. Wang C et al. proposed the use of SB431542 inhibitor as a blocking strategy for TFG-β receptor I signalling [54]. By using OSI906 or PPP compounds specific inhibitors for IGF-1 receptors, Xu J et al. demonstrated that MSC-derived human embryonic stem cells increased the circulating levels of IGF-1 that ultimately maintained the integrity of intestinal epithelia cells and contributed to their repair and regeneration [55]. Interestingly, Kang J et al. [56] investigated the role of secreted miR148b-5p by UC-MSCs in modulating experimental colitis. In this sense, downmodulation of 15-lox-1 expression by macrophages was less effective when colitic mice were treated with miR148b-5p inhibitors, thus confirming the key role of macrophages in modulating in vivo responses. More recently, Tian J et al. demonstrated by RNA interference to beclin (BECN)1 the autophagy pathway is also involved in the therapeutic effects of MSCs [57] (Table 1, Table 2 and Table 7, and Appendix A).

## 4. Strategies to Boost MSC-Based Therapy in IBD

Numerous studies aiming to improve the efficacy of MSC therapy have been conducted in preclinical models of IBD, mainly due the limited beneficial effects observed in the clinical trials conducted to date [6].

### 4.1. Culture Media for MSC Expansion

As depicted in Table 1, different culture media without foetal bovine serum (FBS) have been used by Kang JY et al. [58], Wu X et al. [59], and Liu Y et al. [60] during the ex vivo expansion of the MSCs. These authors observed enhanced immunomodulatory effects of MSCs in in vivo models of IBD, thus reducing the risk of xenoimmunization and zoonotic transmission. In this vein, other investigators have used culture media supplemented with different compounds such as aspirin [61], a combination of b-fibroblast growth factor (FGF), all-trans-retinoic acid (ATRA), and modified neuronal medium (MNM) [62], or activin A and FGF2 [63] and platelet lysates [64]. In these instances, improved in vivo beneficial effects of MSC-based therapy were reported.

In a clinical trial conducted by Dhere et al. [65] fibrinogen-depleted human platelet lysate-supplemented media was used during the ex vivo expansion of the MSCs. In this preliminary phase I clinical trial, the authors reported that autologous BM-derived MSCs expanded under these conditions were well tolerated in patients with medically refractory moderate to severe Crohn’s disease, although no sustained efficacy was observed (Appendix A).

On other hand, it has been amply described that adding pro-inflammatory cytokines such as IFN-γ [46,66,67,68,69], TNF-α [46,68,70], IL-1β [44,71], or a combination of these [40,72,73,74] can increase the immunosuppressive functionality of the treated MSCs. Additionally, the immunoregulatory effects of the MSCs can be enhanced by using pol I:C [46,75,76] alone or in combination with IFN-γ [77], or *l**ipopolysaccharide* (LPS) alone [46] or with adenosine triphosphate (ATP) [78]. These molecules carry out their effects by activation of the Toll-like receptor (TLR) pathways that ultimately activate NACHT, LRR, and PYD domains containing protein (NLRP3) inflammasome. Other molecules such as IL-25 (a member of the cytokine IL-17 family [79]) and thapsigargin (an endoplasmic reticulum stress inducer) showed increased yield and expression of immunomodulatory factors such as TGFβ, COX2, and IDO [80] and muramyl dipeptide that increased the production of PGE_2_ via the NOD2–RIP2 pathway [53]. Interestingly, Xiao E et al. [81] observed that normal microbiota is required to maintain immunomodulatory properties of MSCs. Thus, under germ-free (GF) environments, the stemness of MSCs was altered in comparison to specific-pathogen-free (SPF)-derived MSCs. Colonization of GF mice with SPF microbiota (conventionalized) increased the immunomodulatory capacity of SPF-derived MSCs in a mouse model of colitis. These results may explain, in part, the differences observed in the functionality of expanded MSCs derived from different donors.

**Table 1 biomedicines-09-01507-t001:** Preclinical studies with MSC-base therapy in IBD using different pre-treatments and culture media for MSC expansion (listed in alphabetical order).

Reference	Animal Model	Specie, Strain, Gender	MHC Context	Source of MSCs	Administration Route	MSC Dose	No. of MSCs Infusions	Culture Condition for MSC Expansion	Day of Infusion (D)	Parameters Analysed for IBD Progression	Therapeutic Effect of MSC Therapy
Ahn JS, 2020 [78]	7-day DSS cycle	Mouse, C57/BL6, M	X	UC	IP	2 × 10^6^	1	-WT MSCs -LPS + ATP-pretreated MSCs	D1	-DAI (Rectal bleeding, stool consistency, and general activity)	Yes, LPS + ATP-pretreated MSCs > WT MSCs
An JH, 2020a [74]	7-day DSS cycle	Mouse, C57/BL6, M	X	AD	IP	−100 µg	3	-EVs from WT MSCs -EVs from TNFα and IFNγ-pretreated MSCs	D1, D3, and D5	-DAI (Body weight, general activity, stool consistency, andrectal bleeding) -Colon length and H/E staining -Arg, CD11b, CD11c CD206, COX-2, FOXP3, IFNγ, IL6, IL10, IL17 iNOS, TGFβ, TNFα, and TSG-6 (colon)	Yes, EVs from TNFα + IFNγ-pretreated MSCs > EVs from WT MSCs
Chen Y, 2015 [67]	7-day DSS cycle	Mouse, C57BL/6, F	X	UC	IV	1 × 10^6^	1	-WT MSCs -IFNγ-MSCs -IFNγ-pretreated MSCS	D8	-DAI (Body weight, stool consistency, and rectal bleeding) -Colon H/E staining -Th1, Th2, Th17, and Treg cells (mLN and SP)	Yes, IFNγ-MSCs > IFNγ-pretreated MSCs > WT MSCs
Chen Q, 2019 [82]	TNBS	Mouse, BALBC, F	S	BM	IV	1 × 10^6^	1	-WT MSCs -AntiVCAM1-treated MSCs	24 h	-DAI (Body weight, stool consistency, and rectal bleeding) -Colon H/E staining and IHC (Ki67 and claudin 1) -Th1, Th2, Th17, and Treg (colon)	Yes, antiVCAM1-treated MSCs > WT MSCs
Cheng W, 2017 [79]	7-day DSS cycle	Rat, Sprague Dawley, NA	S	BM	IV	5 × 10^6^	3	-WT MSCs -IL25-pretreated MSCs -Mesalazine	D1, D2, and D3	-DAI (Body weight) -Colon length, H/E staining, and IHC (Ki67 and LGR5) -Th1, Th2, Th17, and Treg cells (PB)	Yes, Mesalazine > IL25-treated MSCs > WT MSCs
Duijvestein M, 2011 [66]	-7-day DSS cycle -TNBS	-Mouse, C57BL/6JIco, F -Mouse, BALB/C, NA	-X -A	BM	IP	−0.5 × 10^6^ −1 × 10^6^	1	-WT MSCs -IFNγ-pretreated MSCs	-D0 -6 h	-DAI (Body weight, stool consistency, and survival) -Colon weight, H/E staining and IHC (CD3) -IFNγ, IL-6, IL10, IL17A, and TNFα (colon) -SAA (serum)	Yes, IFNγ-pretreated MSCs > WT MSCs
Fan H, 2012 [44]	7-day DSS cycle	Mouse, C57/BL6, M	X	UC	IV	2 × 10^6^	1	-WT MSCs -IL1β-pretreated MSCs -IL1R1^−/−^ MSCs	D1	-Body weight, stool consistency, and rectal bleeding -Colon length, weight, and H/E staining -M1 and M2 macrophages (PC) -Th1, Th2, Th17, and Tregs (SP and mLNs)	Yes, IL1β-pretreated MSCs > IL1R1^−/−^ MSCs> WT MSCs
Fuenzalida P, 2016 [75]	7-day DSS cycle	Mouse, C57BL6, NA	X	UC	IP	1 × 10^6^	2	-WT MSCs -Pol I:C-pretreated MSCs -LPS-pretreated MSCs	D1 and D3	-DAI (Body weight, stool consistency, and rectal bleeding) -Colon length and H/E staining	-Yes, Poly (I:C)-pretreated MSCs > WT MSCs -No, LPS-pretreated MSCs
Giri J, 2020 [51]	6-day DSS cycle	-Mouse, C57BL6, F -Mouse, IL10 KO, F	S	BM	IP	1 × 10^7^	2	-WT MSCs -IFNγ-pretreated MSCs -CCL2^−/−^ MSCs	D2 and D4	-DAI (Body weight) -Colon H/E staining	-Yes, IFNγ-pretreated MSCs > WT MSCs -No, CCL2^−/−^ MSCs
Hu S, 2019 [72]	5-day DSS cycle	Mouse, BALB/C, M	NA	NA	IV	1 × 10^6^	1	-WT MSCs -IFNγ + TNFα-pretreated MSCs	D3	-DAI (Body weight, stool consistency, and rectal bleeding) -MPO activity (colon) -IFNγ, IL1β, IL10, IL17, and TNFα, (colon and serum)	-Yes, WT MSCs -No, IFNγ and TNFα-pretreated MSCs
Joo H, 2021 [80]	7-day DSS cycle	Mouse, C57/BL6, NA	X	UC (WJ)	IP	200 µg 200 µL	3	-CM from GW4869-pretreated MSCs -EVs -EVs from TSG-pretreated MSCs	D1, D3, and D5	-Body weight, stool consistency, and rectal bleeding -Colon H/E staining and MPO activity -Arg-1, CD206, F4/80, and CD11b (colon)	-Yes, EVs from TSG-pretreated MSCs > EVs -No, CM from GW4869-pretreated MSCs
Kang JY, 2020 [58]	7-day DSS cycle	Mouse, NA, NA	X	P (WJ)	IP	2 × 10^6^	1	-FBS-pretreated MSCs -XF-pretreated MSCs	D1	-DAI (Body weight, stool consistency, rectal bleeding, coat roughness, activity and survival) -Colon length and H/E staining	Yes, XF-pretreated MSCs > FBS-pretreated MSCs
Kim HS, 2013 [53]	-7-day DSS cycle -TNBS	-Mouse, C57BL/6, NA -Mouse, BALBC, NA	X	UC	IP	2 × 10^6^	1	-WT MSCs -MDP-pretreated MSCs -MDP + siNOD2-pretreated MSCs	-D2 -6 h	-DAI (Body weight, stool consistency, rectal bleeding, coat roughness, and survival) -Colon length, H/E staining, and MPO activity -IL6, TNFα, IFNγ, IL10, CD4, CD11b, Treg, and PGE2 (colon) -PGE2 (serum)	-Yes, MDP-pretreated MSCs > WT MSCs -No, MDP + siNOD2-pretreated MSCs
Ko IK, 2010 [83]	7-day DSS cycle	Mouse, C57BL6, NA	S	BM	IV	1 × 10^6^	1	-WT MSCs -MAdCAM-coated MSCs -VCAM-1 Ab-coated MSCs	D2	-Body weight, and survival -Colon length, H/E staining, and IHC (CD3 and Foxp3)	Yes, MAdCAM-coated MSCs = VCAM-1 Ab-coated MSCs > WT MSCs
Li X, 2019 [84]	7-day DSS cycle	Mouse, BALB/C, M	X	Mens	IV	1 × 10^6^	3	-WT MSCs -SDF-1-pretreated MSCs -AMD3100-pretreated MSCs	D2, D5, and D8	-DAI (Body weight, stool consistency, and rectal bleeding) -Colon length and H/E staining -CD4, CD11c^+^MHCII^+^, CD11c^+^CD86^+^, CD11c^+^CD40^+^, CD25, CD68, CD206 Foxp3, and IL4 (SP) -IL4, IL6, IL10, and TNFα (colon)	-Yes, SDF-1-pretreated MSCs > WT MSCs -No, AMD3100-pretreated MSCs
Lim JY, 2021 [77]	5-day DSS cycle	Mouse, C57/BL6, F	S	BM	IP	3 × 10^6^	2	-WT MSCs -Poly (I:C) + IFNγ-pretreated MSCs	D1 and D3	-DAI (Body weight, stool consistency, and rectal bleeding) -Colon H/E staining and IHC (IDO19, Ki67, and lysozyme) -Alpi, Axin2, Bmi, CD4, CD11b, CD11c, CD25 Chga, COX, Foxp3, IDO1, IL1β, IL10, IL16, LGR5, MCP1, Muc2, OLFM4, PTEGS3, and TNFα, (colon)	Yes, Poly (I:C) and IFNγ-pretreated MSCs> WT MSCs
Liu Y, 2020 [60]	TNBS	Mouse, BALBC, M	X	UC	IP	4 × 10^6^	1	-FBS-pretreated WT MSCs -FS-pretreated WT MSCs -UltraGRO-Advanced MSCs	10 h	-IDO, IL1β, IL6, and PGE2 (serum)	Yes, UltraGRO-Advanced MSCs > FBS-pretreated MSCs = FS-pretreated MSCs
Qiu Y, 2017 [46]	TNBS	Mouse, BALBC, NA	-X -S	UC	IP	1 × 10^6^	1	-WT MSCs -Pol I:C-pretreated MSCs -TNFα-pretreated MSCs -IFNγ-pretreated MSCs -LPS-pretreated MSCs -WT-MSCs + DAPT -TLR3^−/−^ MSCs -Jagged-1^−/−^ MSCs -Notch-1^−/−^ MSCs	-2 h -D4	-DAI (Body weight, stool consistency, and survival) -Colon H/E staining and MPO activity -COX2, IFNγ, IL4, IL6, IL10, IL17A, IL21, IL23, PGE2, and TNFα (colon and serum) -Th1, Th17, and Treg (mLN and SP)	-Yes, Poly (I:C)-pretreated MSCs > IFNγ-pretreated MSCs > LPS-pretreated MSCs = TNFα-pretreated MSCs= WT MSCs -Yes, S = X -Yes, 12 h and D4 -No, WT-MSCs + DAPT, TLR3^−/−^ MSCs, Jagged-1^−/−^ MSCs and Notch-1^−/−^ MSCs
Ryu DB, 2016 [76]	7-day DSS cycle	Mouse, C57BL6, F	S	BM	-IV -IP	1 × 10^6^	1	-WT MSCs -Pol I:C-pretreated MSCs	D3	- DAI (Body weight) -Colon length -IDO (colon)	No, Poly (I:C)-pretreated MSCs = WT MSCs
Salmenkari H, 2019 [64]	6-day DSS cycle	Mouse, BALB/C, M	X	BM	-IV -IP	0.5 × 10^6^	2	-Fresh WT MSCs -Cryopreserved platelet-lysate-expanded MSCs	D3 and D5	-Colon length, H/E staining, and IHC (ACE, corticosterone, IL1-β, and TNFα) -Agtr1a, IL1β, and S18 (colon)	No, Fresh and cryopreserved MSCs
Shin TH, 2020 [68]	7-day DSS cycle	Mouse, C57BL6, M	X	T	IP	2 × 10^6^	1	-WT MSCs -TNFα-pretreated MSCs -IFNγ-pretreated MSCs	-D1 -D5	-DAI (Body weight, stool consistency, rectal bleeding, coat roughness, general activity, and survival) -Colon length and H/E staining	Yes, TNFα-pretreated MSCs > IFNγ-pretreated MSCs > WT MSCs
Song WJ, 2019 [70]	-7-day DSS cycle -DNBS	Mouse, C57BL6, M	X	AD	IP	2 × 10^6^	1	-WT MSCs -TNFα-pretreated MSCs	-D1 -6 h	-DAI (Body weight, rectal bleeding, and general activity) -Colon H/E staining -CD206, F4/80, IL1β, IL6, IL10, and iNOS (colon)	Yes, TNFα-pretreated MSCs > WT MSCs
Tang, J, 2014 [61]	10-day DSS cycle	Mouse, C57BL6, F	S	BM	IV	1 × 10^6^	1	-WT MSCs -ASA-pretreated MSCs	D3	-DAI (Body weight, stool consistency, and rectal bleeding) -Colon H/E staining and IHC (IL17 and Treg) -IL10, IL17, and Treg (serum)	Yes, ASA-pretreated MSCs > WT MSCs
Tian J, 2020 [57]	10-day DSS cycle	Mouse, C57BL6, NA	S	BM	IV	0.5 × 10^6^	1	-WT MSCs -Pitsopt2-pretreated MSCs -Colivelin-pretreated MSCs -siBECN1-MSCs	D3	-DAI (Body weight, stool consistency, and rectal bleeding) -Colon length and H/E staining -Th17 and Treg (SP) -BECN1, CD4, CD8, CD9, CD63, and CD81 cells, corticosterone, IFNγ, LC3-I, LC3-II, pSTAT3, STAT3, and TNFα, (PB)	-Yes, WT MSCs > colivelin-pretreated MSCS -No, Pitsopt2-pretreated MSCs and siBECN1-MSCs
Wu X, 2020 [59]	7-day DSS cycle	Mouse, C57BL6, M	X	UC	IP	1 × 10^6^	−1	-WT MSCs -Serum free MSCs	D7	-DAI (Body weight, stool consistency, and rectal bleeding) -Colon length and H/E staining - F4/80, CD45, CD86, and CD206 (colon, PB and SP) - Arg1, IL4, IL10, iNOS, MCP1, and TNFα (colon)	Yes, Serum free MSCs > WT MSCs
Xiao E, 2017 [81]	9-day DSS cycle	Mouse, C57BL6, NA	S	BM	IV	1 × 10^6^	1	-WT MSCs form SPF mice -WT MSCs from GF mice -WT MSCs from ConvD mice	D3	-DAI (Body weight, stool consistency and rectal bleeding) -Colon H/E staining	-Yes, WT MSCs from SPF = WT MSCs from ConvD -No, WT MSCs from GF mice
Yang FY, 2018 [62]	6-day DSS cycle	Mouse, C57BL6, F	X	UC	IP	2 × 10^6^	2	-WT MSCs -b-FGF, ATRA + MNM-pretreated MSCs	D2 and D6	-DAI (Body weight and stool consistency) -Colon H/E staining, IHC (Ly-6B.2, occludin1, and ZO-1), and Tunnel analysis -Bcl2, CD3, cleaved-caspase 3, PCNA, p21, p44/42, p65, and pJnk (colon)	Yes, bFGF, ATRA and MNM pretreated MSCs > WT MSCs
Yang R, 2018 [45]	10-day DSS cycle	Mouse, C57BL6, NA	S	G	IV	0.2 × 10^6^	1	-WT MSCs -NaHS-pretreated Cbs^−/−^ MSCs -Cbs^−/−^ MSCs	D3	-DAI (Body weight and stool consistency) -Colon H/E staining -Th17 and Treg (colon)	-Yes, WT MSCs = NaHS-pretreated Cbs^−/−^ MSCs -No, Cbs^−/−^ MSCs
Ye L, 2018 [63]	7-day DSS cycle	Mouse, BALB/C, M	X	BM	IR	1 × 10^6^	2	-Activin A + FGF2-pretreated MSCs	D7 and D9	-Body weight -Colon length and H/E staining -IFNγ, IL6, IL10, and TNFα (colon)	Yes
Yu D, 2021 [71]	7-day DSS cycle	Mouse, BALB/C, M	X	E	IV	1 × 10^6^	3	-WT MSCs -IL1β-pretreated MSCs	D2, D5 and D8	-DAI (Body weight, stool consistency, and rectal bleeding) -Colon length, H/E staining, and IHC (CD206 and iNOS) -β-catenin, DCs, IFNγ, IL4, IL6, IL10, IL17, M2 macrophages, Th1, Th2, Th17, Treg, and TNFα (colon) -β-catenin (SP)	Yes, IL1β-pretreated MSCs > WT MSCs
Yu Y, 2019 [73]	7-day DSS cycle	Mouse, C57BL6, M	X	UC	IV	1 × 10^6^	1	-WT MSCs -IFNγ + IL1β-pretreated MSCs	D2	-Body weight -Colon length and H/E staining -IL6 (serum)	-Yes, IFNγ + IL1β-pretreated MSCs -No, WT MSCs
Zhang S, 2021 [40]	7-day DSS cycle	Mouse, C57BL6, NA	X	Musc	IV	2.5 × 10^5^	1	-WT MSCs -IFNγ and TNFα pretreated MSCs -IDO^−/−^ MSCs -TSG6^−/−^ MSCs -TSG6 -IDO^−/−^ MSCs + KYN -IDO^−/−^ MSCs +KYNA	D2	-Body weight -Colon H/E staining -IL6 (serum)	Yes, WT MSCs = TSG6 > IDO^−/−^ MSCs + TSG6 = IDO^−/−^ MSCs + KYN = IDO^−/−^ MSCs +KYNA > IDO^−/−^ MSCs = TSG6^−/−^ MSCs

Therapeutic effect; > better than, =similar to; < less than.

Additionally, some groups have proposed to increase the therapeutic effect of MSCs by enhancing their in vivo trafficking to the inflammation site. In line with this, Chen Q et al. [82] and Ko IK et al. [83] incubated MSCs with antibodies against vascular cell adhesion molecule (VCAM1). The authors observed increased therapeutic effects of the VCAM1-treated MSCs thanks to their increased migration to the injured colon. Li X et al. [84] pretreated MSCs with stromal derived factor 1 (SDF-1) and, as a consequence, increased expression of CXCR4 chemokine receptor was induced. Interestingly, SDF-1-treated MSCs were found to engraft to injured colons at a greater rate than native MSCs, thus increasing their therapeutic effect (Table 1 and Table 6).

### 4.2. MSC Therapy in Combination with Other Molecules

Several investigators have used MSC-based therapy in combination with different molecules (Table 2) used as traditional anti-inflammatory treatments such as sulfasalazine [85] or even novel treatments such as those derived from microbiota in IBD experimental animal models to increase the beneficial effects of MSC therapy. In this line, Lee BC et al. [86] combined MSC-based therapy with MIS416, a novel microparticle derived from *Propionibacterium acnes* that activates NOD2 and TLR9 pathways. Mar JS et al. [87] combined MSC therapy with a microbial supplement (VSL#3) to re-establish appropriate microbiota composition in the ileum. These combined approaches have shown to increase the therapeutic effects of MSCs. On the other hand, Simovic-Markovic B et al. observed that galectin-3 inhibitor enhanced MSC therapy’s potential to attenuate DSS-induced colitis by increasing IL-10 concentration in sera of DSS-treated animals due to the polarization towards immunosuppressive M2 phenotype of macrophages [88].

Tang Y et al. observed that MSC therapy in combination with granulocyte colony-stimulating factor (G-CSF) administration enhanced the therapeutic effect of MSCs with respect to either alone. G-CSF is currently the most widely used stem cell mobilization agent, which can induce spontaneous stem cell homing to damaged tissue and proliferation of cells in the damaged tissue thus promoting tissue repair [89]. In this line, Forte D et al. [90] used platelet lysates as a vehicle to infuse MSCs as it is known that platelet-rich plasma accelerated wound healing and bone regeneration. Chang CL et al. observed reduced colon permeability and injury scores as well as a decrease in inflammatory markers when MSC-derived exosomes were used in combination with *N*-acetyl-5-methoxytryptamine (melatonin) [91], a hormone mainly produced by the mitochondria of the pineal gland that exerts anti-oxidative effects and ameliorates damage caused by reactive oxygen and nitrogen species [92].

Several studies have documented that expanded MSCs have procoagulant activity that could induce disseminated coagulation and thrombosis in recipients. Anticoagulation treatment using heparin is a practical strategy to improve both the safety and therapeutic effect of MSC therapy as proposed by Liao L et al. By the use of this strategy, the authors demonstrated that the total amount of MSCs that can be infused into mice could be increased up to four times with no risk of disseminated intravascular coagulation. Additionally, heparin treatment improved the distribution of MSCs to the DSS-inflamed colons in colitic mice [93].

**Table 2 biomedicines-09-01507-t002:** Preclinical studies of MSC-based therapy in IBD in combination with other treatments (listed in alphabetical order).

Reference	Animal Model	Specie, Strain, Gender	MHC Context	Source of MSCs	Administration Route	MSC Dose	No. of MSCs Infusions	Treatments Used	Day of Infusion (D)	Parameters Analysed for IBD Progression	Therapeutic Effect of MSC Therapy
Chang CL, 2019 [91]	5-day DSS cycle	Rat, NA, NA	A	AD	NA	50 µM/Kg	3	-Evs -Evs + Melatonin -Melatonin	D5, D7, and D10	-Colon H/E staining and IHC (CD4, CD14, CD68, COX-2, γCD3 GPx, H2AX, ICAM-1, and NQO1) -Bax, BMP2casp3, COX2, HO-1 ICAM-1, IL1β, IL6, IL10, iNOS, NF-κB, NOX1, NOX2, NOX4, c-PARP, pSmad1/5, pSmad3, TGF-β, TLR4, and TNF-α (colon)	Yes, EVs + Melatonin > EVs > Melatonin
Forte D, 2015 [90]	DSS	Mouse, C57/BL6, M	X	AD	IR	1 × 10^6^	3	-WT MSCs -WT MSCs + Platelet lysate	D7, D9, and D11	-DAI (Body weight, stool consistency, rectal bleeding, and general activity) -Colon H/E staining	Yes, Platelet lysate + MSCs > WT MSCs
Lee BC, 2018 [86]	7-day DSS cycle	Mouse, C57BL6, M	X	UC	IP	2 × 10^6^	1	-WT MSCs -WT MSCs + MIS416 -MIS416	D1	-DAI (Body weight and survival) -Colon length, H/E, PSR staining and MPO activity -IFN-γ, IL-6, IL-10, IL-12p70, MCP-1, and TNFα (serum) -COX2, pERK, IκB-α, IKK-α, iNOS, JNK pJNK, NF-κB p65, pNF-κB, p38, pp38, p65, and RIP2 (colon)	Yes, WT MSCs + MIS416 > WT MSCs = MIS416
Liao L, 2017 [93]	13-day DSS cycle	Mouse, C57/BL6, F	S	BM	IV	1 × 10^6^	1	-WT MSCs -WT MSCs + heparin	D3	-DAI (Body weight and survival) -Colon length and H/E staining -CD3^+^ T cell levels, apoptosis, and proliferation (colon and mLNs)	Yes, MSC + heparin > MSCs
Park HJ, 2018 [52]	2 × 5-day DSS cycle	Mouse, C57BL6, F	X	AD	IP	1 × 10^6^	2	-WT MSCs -WT MSCs + celecoxib	D6 and D16	-Body weight -Colon length, H/E staining, and IHC (F4/80)	Yes, WT MSCs + celecoxib > WT MSCs
Simovic Markovic B, 2016 [88]	6-day DSS cycle	-Mouse, C57BL6, M -Mouse, Gal-3 KO, M	S	BM	IP	0.5 × 10^6^	2	-WT MSCs -WT MSCs + Gal3 inhibitor	D0 and 12 h	-DAI (Body weight, stool consistency, and rectal bleeding) -Colon H/E staining -Gal-3, IL-1*β*, IL-10, TGF-*β*, and TNF-α (serum) -CD, CD11b, CD11c, CD117, CD206, CD80, F4/80, and NK 1.1 (colon) -IL-1*β,* IL-10, IL-12 and TNF-*α (colon)*	Yes, WT MSCs = WT MSCs + Gal-3 inhibitor
Tang Y, 2015 [89]	TNBS	Rats, Sprague Dawley, NA	S	BM	IV	2 × 10^6^	1	-WT MSCs -5-ASA -WT MSCs + GCSF -GCSF	D1	-DAI (Body weight, stool consistency, and rectal bleeding) -Colon IHC (BrDU and NFκB), H/E staining, and MPO activity -IL10 and TNF-α (serum)	Yes, 5-ASA > WT MSCs + GCSF > WT MSCs = GCSF
Yousefi-Ahmadipour A, 2018 [85]	TNBS	Rats, Wistar, Rats, M	X	AD	IP	1 × 10^6^	2	-WT MSCs -WT MSCs + sulfasalazine -Sulfasalazine	D1 and D5	-DAI (Body weight, stool consistency, and rectal bleeding) -Colon macroscopic analysis -Colon H/E staining and MPO activity -Arg-1, Bax, Bcl2, Foxp3, IL-1, IL-6, IL10, IL17, MCP-1, NFκB, TGFβ, and TNF-α (Colon)	Yes, WT MSCs + sulfasalazine > WT MSCs > sulfasalazine

### 4.3. MSC Therapy in Combination with Alternative Cell Therapies

Considering MSC-based therapy in combination with other cell therapies (Table 3), Abbasi-Kenarsari H et al. showed that the co-transplantation of MSCs and tolerogenic dendritic cells (Tol-DCs) was more effective in alleviating the clinical and histological manifestations of colitis than monotherapy with MSCs [94]. Moreover, Liu X et al. observed amelioration of colitis signs after treatment with a combination of BM-MSCs and HSCs [95], although no comparison with either treatment alone was reported. Additionally, Wei Y et al. observed a reduced colitis score in a TNBS-induced model when MSCs and HSCs were co-infused into colitic mice with respect to either treatment alone [96]. On the other hand, Yun Y et al. observed synergistic effects when a combined cell therapy with MSCs and regulatory T cells (Tregs) was tested with respect to single cell therapy with MSCs or Tregs alone [97].

These results suggest that the combination of MSC therapy with other treatments currently used in the clinic for inflammatory bowel diseases could be a good option to achieve robust therapeutic effects of MSC-based therapy. It must be pointed out that the patients enrolled in the clinical trials for MSC-based therapy maintained their anti-inflammatory treatments during their participation in the trials.

### 4.4. Scaffolding Methods for MSC Therapy

As only a small proportion of systemically infused MSCs reach the inflamed colon [18,22], several groups have developed methods to increase MSC survival and engraftment using different scaffolding methods together with local infusion of the MSCs (Table 4). To this purpose, some authors have infused MSCs locally embedded in spheroids [98,99,100] or cultured them in temperature-responsive culture dishes that can be harvested as intact sheets [101] with similar results as compared to naked MSCs. As an alternative, the use of bioactive hydrogel immobilizing the C-domain peptide of insulin-like growth as scaffold structure has demonstrated better results than free-scaffold MSCs [102]. Recently, Regmi S et al. infused MSCs intraperitoneally in different types of two- or three-dimensional structures formed by polylactic-co-glycolic acid (PLGA) microspheres. They also obtained beneficial effects in different models of colitis, although in these instances a direct comparison to conventional infusion of MSCs was not reported [103,104,105].

Although several investigators have used scaffolding methods for MSC therapy, no significant increase in the therapeutic effects of MSCs has been observed in comparison to non-scaffolded MSCs [98] and in most of the studies no direct comparison to conventional MSCs were reported. In addition, these methods increased the survival of MSCs when local infusion is carried out, thus putting into question the beneficial systemic effects of the MSCs.

**Table 4 biomedicines-09-01507-t004:** Scaffolding methods for MSC-based therapy in IBD (listed in alphabetical order).

Reference	Animal Model	Specie, Strain, Gender	MHC Context	Source of MSCs	Administration Route	MSC Dose	No. of MSCs Infusions	Scaffolding Method	Day of Infusion (D)	Parameters Analysed for IBD Progression	Therapeutic Effect of MSC Therapy
Barnhoorn M, 2018 [98]	7-day DSS cycle	Mouse, C57BL/6Jico, F	S	BM	IR	−2 × 10^6^ −2 × 10^6^ in spheroids	1	-WT MSCs -MSCs in spheroids	D5	-DAI (Body weight and stool consistency) -Colon H/E staining and IHC (Caspase 3, CD3, CD200, CXCR4, Foxp3, IDO, Ki67, Ly6G, and MBP) -SAA, TGF-β1, and VEGF (colon)	Yes, WT MSCs = MSCs in spheroids
Cao X, 2020 [102]	TNBS	Mouse, BALB/C, M	X	P	IR	1 × 10^6^	1	-WT MSCs -CS Hydrogel MSCs -CS-IGF-IC Hydrogel MSCs	24 h	-DAI (Body weight) -Colon length, H/E and MPO staining, MPO activity, and IHC (CD206, EpCAM F4/80, GFP iNOS, and PGE2,) -IFNγ, IL1β, IL6, and TNFα (colon)	Yes, CS-IGF-IC Hydrogel MSCs > CS Hydrogel MSCs > WT MSCs
Molendijk I, 2016 [100]	7-day DSS cycle	Mouse, C57BL/6Jico, F	S	BM	IR	−0.5 × 10^6^ in spheroids −2 × 10^6^ in spheroids	1	MSCs in spheroids	D5	-DAI (Body weight) Colon H/E staining, IHC (F4/80 and FOXP3), and MPO activity -SAA (serum) -COX2, IFN-ɣ, IL-2, IL-4, IL-6, IL-10, IL-17a PGE2, and TNF-α, (colon)	-Yes, 2 × 10^6^ in spheroids -No, 0.5 x10^6^ in spheroids
Pak S, 2018 [101]	DNBS	Rats, Sprague Dawley, M	S	-AD -BM	IR	−8 × 10^5^ −1.1 × 10^6^ in sheets	1	MSCs in sheets	D4	-DAI (Body weight) -Macroscopy analysis -Colon H/E and eGFP staining	Yes, AD-MSCs and BM-MSCs
Pathak S, 2019 [105]	7-day DSS cycle	Mouse, C57BL/6, NA	X	AD	IP	2 × 10^3^	1	-MSCs in D-PEMA-PLGA spheroids -MSCs in IDN-PLGA spheroids -MSCs in PLGA spheroids	D1	-Body weight and stool consistency -Colon length, MPO activity, and H/E staining -CD4 and CD8 T cells (colon) -INFγ and IL17 (mLN)	Yes, MSCs in IDN-PLGA > MSCs in D-PEMA-PLGA > MSCs in PLGA spheroids
Regmi S, 2020 [104]	7-day DSS cycle	Mouse, C57BL/6, M	X	AD	IP	2 × 10^6^	1	-2D MSCs -3D MSCs -LC3BsiRNA-3D MSCs	D1	-DAI (Body weight, stool consistency, and rectal bleeding) -Colon H/E staining and MPO activity -TGFβ and TNFα (colon)	-Yes, 3D MSCs > 2D MSCs -No, LC3BsiRNA- 3DMSCs
Regmi S, 2021 [103]	7-day DSS cycle	Mouse, C57BL/6, NA	X	AD	IP	2 × 10^6^	1	-2D MSCs -3D MSCs	D1	-DAI (Body weight, stool consistency, and rectal bleeding) -Colon length, H/E staining, and MPO activity -CD4^+^ and CD8^+^ T cells, IFNγ and IL17 (colon)	Yes, 3D MSCs > 2D MSCs
Yan L, 2018 [99]	-7-day DSS -TNBS	-Mouse, C57BL/6, M -Mouse, BALB/C, M	X	E	IP	1 × 10^6^	2	3D-MSCs in spheroids	-D1 and D2 -6 h	-Body weight -Colon length and H/E staining	Yes, DSS and TNBS

Therapeutic effect; > better than, = similar to; < less than.

### 4.5. MSC-Derived Vesicles and MSC-Conditioned Medium

Most of the beneficial effects of MSCs are mediated by paracrine pathways through the secretion of factors packaged in EVs. To confirm these effects of conditioned medium from MSCs (CM-MSCs), Joo H et al. and Tian J et al. observed that culture media from MSCs treated with GW4869 (inhibitor of EV secretion) [80] or with Pistop2 (cell membrane-permeable clathrin inhibitor) [57] inhibited the therapeutic effects of CM-MSCs or MSCs in a DSS-induced colitis model, suggesting that the most important factors responsible for the therapeutic effects of CM-MSCs are packaged and carried out within EVs. The large majority of studies conducted with CM-MSCs have observed similar therapeutic effects to MSCs [106,107,108,109,110] in DSS, as well as in TNBS-induced colitis models regardless of the tissue source, MHC context, or route of administration used (Table 5). Ikarashi S et al. [33] are the only researchers that have reported enhanced therapeutic effects of MSCs when compared to CM-MSCs.

Based on these observations, it can be concluded that EVs from MSCs exert similar beneficial effects to whole MSCs in preclinical models of colitis although, as mentioned earlier, in most of the preclinical studies, no direct comparison to conventional MSC-based therapy has been formally addressed (Table 6).

CM-MSCs and MSC-derived EVs are attractive strategies for MSC-based therapy as they reduce potential immune reactions against MSCs or even tumorigenic potential of living MSC administration. Additionally, they can be concentrated, frozen, or even lyophilized without loss of activity, which gives them a certain advantage over whole MSCs that require liquid nitrogen storage and an adequate infrastructure to preserve and revive frozen cells. Moreover, EVs can be loaded with a great variety of proteins, peptides, RNA, and lipid mediators. Despite these advantages, more studies are necessary, as well as direct comparison with conventional MSC-based therapy.

**Table 5 biomedicines-09-01507-t005:** Preclinical studies of IBD using MSC-conditioned media (listed in alphabetical order).

Reference	Animal Model	Specie, Strain, Gender	MHC Context	Source of MSCs	Administration Route	MSC Dosage/ Volume of CM-MSC	No. of MSCs Infusions	Type of Conditioned Media	Day of Infusion (D)	Parameters Analysed for IBD Progression	Therapeutic Effect of MSC Therapy
Heidari M, 2018 [108]	3 × 4-day DSS cycle	Mouse, C57/BL6, F	S	AD	IP	−1 × 10^6^ −500 µL	2	-MSCs -CM-MSCs	D16–D27	-Body weight, stool consistency, and rectal bleeding -Colon length, weight, and H/E staining -IL10, IL17, TGFβ, and Tregs (LNs and SP)	Yes, MSCs = CM-MSCs
Ikarashi S, 2019 [33]	7-day DSS cycle	Mouse, C57/BL6, M	X	-UC -AD	IV	−1 × 10^6^ −250 µL	1	-MSCs -CM-MSCs	-D3 -D7	-DAI (Body weight, stool consistency, and rectal bleeding) -Colon length and H/E staining -CCR6, CD19, EGFR, IL6, IL10, IL17, TCR, and TGFβ, VEGF, TNFα, Bacteroidetes, and Firmicutes (Colon) -C10orf54, CA9, CLGN, *C*XCL5, DHRS3, FHOD3, IL-6, KRT7, RARRES1, RRAD, STC1, and TNFRSF11B (Serum*)*	-Yes, D3 -Yes, UC-MSCs = AD-MSCs > CM-MSCs -No, D7
Joo H, 2021 [80]	7-day DSS cycle	Mouse, C57/BL6, NA	X	UC (WJ)	IP	−200 µg −200 µ	3	-CM from GW4869-pretreated MSCs -EVs from WT MSCs -EVs from Thapsigargin-pretreated MSCs	D1, D3, and D5	-Body weight, stool consistency, and rectal bleeding -Colon H/E staining and MPO activity -Arg-1, CD11b, CD206, and F4/80 (colon)	-Yes, EVs from TSG-pretreated MSCs > EVs -No, CM from GW4869-pretreated MSCs
Lee KE, 2019 [110]	5-day DSS cycle	Mouse, C57BL6, M	X	T	IP	−1 × 10^6^ −500 µL	−4 −12	-MSCs -CM-MSCs	-D6, D10, D12, and D16 -D6 every 2 days	-DAI (Body weight) -Colon length and H/E staining -IL1β, IL6, IL10, IL17, and TNFα (colon)	Yes, MSCs = CM-MSCs
Legaki E, 2016 [111]	5-day DSS cycle	Mouse, NOD/SCID, NA	X	Amniotic fluid	IP	200 µL	1	CM-MSCs	D5	-Body weight, stool consistency, and rectal bleeding -Colon H/E and IHC (MMP1, MMP2, TGF-β, and TNFα) staining -IL1β, IL10, and TNFα (colon)	Yes
Lykov AP, 2018 [109]	7-day DSS cycle	Mouse, C57BL6, M	S	BM	IV	−2 × 10^5^ −200 µL	1	-MSCs -CM-MSCs	D8	-Small intestine H/E staining	Yes, MSCs = CM-MSCs
Miyamoto S, 2017 [112]	TNBS	Rats, Sprague Dawley, M	X	Amniotic fluid	-IV -IR	400 µL	1	CM-MSCs	-3 h -3, 24, and 48 h	-Endoscopic analysis -Colon H/E and MPO staining -CD3, CD68, CXCL1, CCL2, IL6, and TNFα (colon)	Yes
Pouya S, 2018 [113]	4-day DSS cycle	Mouse, C57BL6, F	S	AD	IP	500 µL	3	CM-MSCs	D4, D6, and D8	-DAI (Body weight, stool consistency, rectal bleeding, and survival) -Colon length and H/E staining -IL10, IL17, TGFβ, and Treg (mLN and SP)	Yes
Robinson AM, 2014 [106]	TNBS	Pigs, Guinea, M and F	X	BM	IR	−1 × 10^6^ −300 µL	1	-MSCs -CM-MSCs	3 h	-Body weight -Colon H/E and Alcian blue staining and IHC (CD45, ChAT, GFP, nNOS, PGP9.5, and pan-neuronal) -Colonic motility	Yes, MSCs = CM-MSCs
Robinson AM, 2015 [107]	TNBS	Pigs, Guinea, M and F	X	BM	IR	−1 × 10^6^ −300 µL	1	-MSCs -CM-MSCs	3 h	-Body weight -Colon H/E and Alcian blue staining and IHC (CD45, ChAT, GFP, HLA-A, HLA-B, HLA-C, nNOS, PGP9.5, and pan-neuronal) -Colonic motility	Yes, MSCs = CM-MSCs
Watanabe S, 2013 [114]	-6-day DSS cycle -TNBS	Rat, NA, NA	S	NA	-IV -IP -IR	−200 µL −500 µL	−3 −5	-CM MSC	-D3–D 5-D3–D7	-DAI -Colon H/E staining and IHC (Ki67)	-Yes, DSS and TNBS models -Yes, 500 µL × 5> 500µL × 3 = 200 µL × 3 -Yes, IP = IV = IR

**Table 6 biomedicines-09-01507-t006:** Preclinical studies of IBD using MSC-derived vesicles (listed in alphabetical order).

Reference	Animal Model	Gender, Strain, Specie	MHC Context	Source of MSCs	Administration Route	MSC Dose	No. of Infusions	Type of EV-Derived MSCs	Day of Infusion (D)	Parameters Analysed for IBD Progression	Therapeutic Effect of MSC Therapy
An JH, 2020b [43]	7-day DSS cycle	Mouse, C57/BL6, M	X	AD	IP	100 µg	3	-EVs from WT MSCs -EVs from TSG-6^−/−^ MSCs	D1, D3, and D5	-DAI (Body weight, stool consistency, rectal bleeding, and general activity) -Colon length and H/E staining -Arg, CD206, COX-2, CD11b, CD11c FOXP3, IFNγ IL6, IL10, IL17, iNOS, TGF-β, TSG-6, and TNF-α (colon)	-Yes, EVs from WT MSCs -No, EVs from TSG-6^−/−^ MSCs
An JH, 2020a [74]	7-day DSS cycle	Mouse, C57/BL6, M	X	AD	IP	100 µg	3	-EVs from WT MSCs -EVs from TNFα + IFNγ-pretreated MSCs	D1, D3, and D5	-DAI (Body weight, stool consistency, rectal bleeding, and general activity) -Colon length and H/E staining -Arg, CD206, COX-2, CD11b, CD11c FOXP3, IFNγ IL6, IL10, IL17, iNOS, TGF-β, TSG-6, and TNF-α (colon)	Yes, EVs from TNFα + IFNγ-pretreated MSCs > EVs from WT MSCs
Cao L, 2019 [115]	7-day DSS cycle	Mouse, BALB/C, M	S	BM	IP	50 µg	7	EVs from WT-MSCs	D0–D7	-DAI (Body weight, stool consistency, and rectal bleeding) -Colon macroscopic analysis, H/E staining, and MPO activity -Colon IHC (CD86 and CD163) -CD163, pJAK1, JAK1, pSTAT1, and pSTAT6 (Colon)	Yes
Chang CL, 2019 [91]	5-day DSS cycle	Rat, NA, NA	A	AD	NA	50 µM/Kg	3	-EVs from WT MSCs-EVs + Melatonin -Melatonin	D5, D7, and D10	-Colon H/E staining and IHC (CD3, CD4, CD14, CD68, COX-2, GPx, ICAM-1, γ-H2AX, and NQO1) -Bax, BMP2 casp3, COX2, IL1β, IL6, IL10, HO-1, ICAM-1, iNOS, NF-κB, NOX1, NOX2, NOX4, TLR4, c-PARP, pSamd3, psmad1/5, TGF-β, and TNF-α (colon)	Yes, EVs + Melatonin > Evs from WT MSCs > Melatonin
Chen Q, 2020 [116]	TNBS	Rat, Sprague Dawley, M	S	BM	IV	−50 µg −100 µg −200 µg	1	EVs from WT MSCs	D3	-DAI (Body weight, stool consistency, and rectal bleeding) -Colon H/E staining -CD4 and IL17 (mLN and SP) -EED, EZH1, EZH2, IL17, IL21, IL22, jmjd3 RORγT, STAT3, SUZ12, and UTX (colon)	Yes, 100 µg = 200 µg > 50 µg
Duan L, 2020 [117]	TNBS	Mouse, BALB/C, M	X	P	IR	200 µg	1	EVs from WT MSCs	24 h	-Body weight -Colon H/E staining and IHC (EpCAM, Ki67, MPO, and ROS) -casp3, casp8, casp9, IFNγ, IL1β, IL6, IL10, TGFβ, and TNFα (colon)	Yes
Heidari M, 2021 [118]	7-day DSS cycle	Mouse, C57/BL6, F	S	AD	IP	100 µg	3	EVs from WT MSCs	D2, D4, and D6	-DAI (Body weight, stool consistency, and rectal bleeding) -Colon length and weight, H/E staining and MPO activity -IFN-γ, IL-4, IL-10, IL-12, IL-17, TGF-β, TNF-α, and Treg (mLN and SP)	Yes
Joo H, 2021 [80]	7-day DSS cycle	Mouse, C57/BL6, NA	X	UC (WJ)	IP	−200 µg −200 µL	3	-CM from GW4869-pretreated MSCs -EVs from WT MSCs -EVs from TSG-pre-treated MSCs	D1, D3, and D5	-Body weight, stool consistency and rectal bleeding -Colon H/E staining and MPO activity -Arg-1, CD11b, CD206, and F4/80 (colon)	-Yes, EVs from TSG-pretreated MSCs > Evs from WT MSCs -No, CM from GW4869-pretreated MSCs
Li Y, 2020a [119]	8-day DSS cycle	Mouse, C57BL6, NA	X	AD	IP	1 × 10^7^	1	-WT MSCs -Evs from WT MSCs	D4	-Body weight, stool consistency, and rectal bleeding -Colon H/E staining -IFNγ, IL6, IL10, IL12, IL17, and TNFα (colon) -JAK2, JNK, and STAT3 (colon)	Yes, EVs from WT MSCs = WT MSCs
Liu H, 2019 [35]	-TNBS -7-day DSS cycle	-Mouse, BALB/C, M -Mouse, C57/BL6, M	X	BM	IV	−30 µg −100 µg −200 µg −400 µg	−1 −2 −3	EVs from WT MSCs	-D7 -D7 and D16 -D2 -D2 and D4 -D2, D4, and D6	-DAI (Body weight, stool consistency, and rectal bleeding) -Colon length, H/E staining, and MPO activity -Ang4, Arg1, CD11b, Defa20, Defa29, F4/80, IFN-γ, IL-1β, IL-6, IL-10, Lyz1, Rentl1a, TNF-α, Th,1 and Th2 (colon)	-Yes, TNBS and DSS -Yes, D7and D16 > D7 -Yes, D2, D4, and D6 = D2 and D4 > D2
Ma ZJ, 2019 [120]	6-day DSS cycle	Mouse, C57BL6, NA	X	UC	IP	−200 µg −1 × 10^6^	1	-WT MSCs -EVs from WT MSCs	D2	-DAI (Body weight, stool consistency, and rectal bleeding) -Colon length and H/E staining -IFNγ, IL6, IL10, IL17, TGFβ, and TNFα (colon)	Yes, EVs > WT MSCs from WT MSCs
Mao F, 2017a [121]	11-day DSS cycle	Mouse, KM, M	X	UC	IV	−1.3 × 10^6^ −400 µg	3	-WT MSCs -EVs from WT MSCs	D3, D6, and D9	-Body weight -Colon length, H/E staining, and IHC (CD206, IL7, and PCNA) -IL1β, IL6, IL10, IL7, iNOS, and TNFα (colon and SP)	Yes, WT MSCs = EVs from WT MSCs
Song J, 2017 [122]	-7-day DSS cycle -3 × 5-day DSS cycle	Mouse, C57BL6, NA	X	UC	IP	150 µg	−1 −5	EVs from WT MSCs	-D3, D4, D5, D6, and D7 -D26	-DAI (Body weight, stool consistency, and rectal bleeding) -Colon MPO activity and IHC (Arg-1, CD68, and iNOS) -Arg-1, CCL1, CXCL9, IL10, IL17, Light, MCP1, and TGFβ (colon)	Yes
Tian J, 2020 [123]	4-day DSS cycle	Mouse, C57BL6, M	S	N	IV	60 µg	2	EVs from WT MSCs	D2 and D4	-DAI (Body weight, stool consistency, and rectal bleeding) -Colon H/E staining	Yes
Tolomeo AM, 2021 [124]	6-day DSS cycle	Mouse, C57BL6, NA	S	BM	IP	4 × 10^6^ 1 × 10^9^	−2 −3	-WT MSCs -EVs from WT MSCs	-D4 and D8 -D4, D6, and D8	-DAI (Body weight, stool consistency, and rectal bleeding) -Colon length, H/E staining, IF (CD45, CD163, Muc5ac, and iNOS) and IHC (Angiopoietin, CD31, and VEGF)	Yes, WT MSCs = EVs from WT MSCs
Wang G, 2020 [125]	DSS	Mouse, BALB/C, M	X	UC	IV	1 mg	3	EVs from WT MSCs	D3, D6, and D9	-DAI (Body weight, stool consistency, and rectal bleeding) -Colon H/E staining -CD9, CD63, cullin 1, DCNL1, E2M, IBC12F, Iκb, IL1β, IL6, IL10, NAe1, NEDD8, NFκB, and pNFκB, PCNA, TNFα, and Uba3 (colon)	Yes
Wu H, 2019 [126]	TNBS	Rat, Sprague Dawley, M	S	BM	IV	100 µg	1	-EVs from WT MSCs -EVs from miR-146a-MSCs	D3	-DAI (Body weight, stool consistency, and rectal bleeding) -Colon length and H/E staining -IκBα, IL-1β, IL-6, IL-10, IRAK1 NF-κBp65, TNF-α, and TRAF6 (colon)	Yes, miR-146a-EVs > EVs from WT MSCs
Wu Y, 2018 [127]	11-day DSS cycle	Mouse, BALB/C, M	X	UC	IV	400 µg	3	EVs from WT MSCs	D3, D6, and D9	-DAI (Body weight, stool consistency, and rectal bleeding) -Colon H/E staining -CD9, CD63, CD81, E2M, FK2, IL1β, IL6 IL10, IP10, K48, K63, Nae1, TNFα, Uba3, and ubiquitin 1 (colon)	Yes
Yang J, 2015 [128]	TNBS	Rat, Sprague Dawley, M	S	BM	IV	−50 µg −100 µg −200 µg	1	EVs from WT MSCs	D3	-DAI (Body weight and stool consistency) -Colon H/E staining and GSH, MDA, MPO and SOD activity, and IHC (casp3, COX2, iNOS, NFκB, and TNFα) - IL1β and IL10 (colon)	Yes, 200 µg > 100µg > 50µg
Yang R, 2020 [69]	10-day DSS cycle	Mouse, C57BL6, F	S	BM	IV	200 µg	1	-EVs from WT MSCs -EVs from IFNγ-pretreated MSCs	D3	-DAI (Body weight, stool consistency, and rectal bleeding) -Colon length and H/E staining -Th17 and Treg (colon)	Yes, EVs from IFNγ pretreated MSCs > EVs from WT MSCs
Yang S, 2021 [39]	-7-day DSS cycle -TNBS	-Mouse, C57BL6, M -Mouse, BALB/C, M	X	UC	IP	200 µg	1	-EVs from WT MSCs -EVs from TSG-6 siRNa-silenced MSCs -TSG6	-D 5-24 h	-DAI (Body weight, stool consistency, rectal bleeding, and survival) -Colon length, FITC, H/E and PAS staining, MPO activity, and IHC (Claudin-1, Occludin, and ZO-1) -Claudin-1, CXCL14, IL-1β, IL-4, IL-11, IL-12 Occludin, TGF-β, TSG6, and ZO-1 (colon) -Th2 and Th17 (mLN and SP)	-Yes, EVs from WT MSCs = TSG6 -No, siTSG-6 MSCs-EVs
Yu T, 2021 [129]	7-day DSS cycle	Rat, Sprague Dawley, M	S	BM	IV	100 µg	1	-EVs from WT MSCs -EVs from EPhB2-MSCs -Mesalazine	D8	-DAI -Colon H/E staining, length, and MPO actitivy -EphB2, eprhin-B1, FOXP3, GSH, IFNγ, IL1β, IL2, IL6, IL10, IL17, MDA, NFκB, pNFκB, occcludin, RORγT, SOD, pSTAT3, STAT3, TFGβ, TNFα, and ZO1 (colon) -Th17 and Tregs (SP)	Yes, EVs from EPhB2-MSCs = mesalazine > EVs from WT MSCs

Therapeutic effect; > better than, = similar to, < less than.

### 4.6. Genetic Modifications of MSCs

Some authors have genetically modified MSCs to express cytokines with immunosuppressive properties such as IL-12p40 subunit [19] as a natural antagonist of IL-12 and IL23, which are two potent pro-inflammatory cytokines required for Th1 and Th17 immune responses, respectively [130,131]. Wang Q et al. proposed the use of genetically modified MSCs expressing IL-37b [132] which are known to inhibit proinflammatory cytokine and chemokine production, neutrophil infiltration, and cellular apoptosis. Two studies published in 2018 have demonstrated that IL-35-expressing MSCs [133,134] and IL-25-expresing MSCs [135] increased the therapeutic effect of the MSC therapy in a DSS-induced colitis model in mice and in a TNBS-induced inflammation in rats. IL-35 is a molecule known to mediate expansion of Tregs, and suppression of Th17 cell differentiation and IL-25 can play a key role in colonic immune tolerance via suppression of mucosal Th1/Th17 responses. Some authors have proposed the use of miRNA technology to regulate gene expression by binding to the 3′-untranslated region (3′-UTR) of target messenger RNAs (mRNAs). Liu L et al. used miR-181a-overexpressed MSCs, as miR-181a regulates T and B cell development as well as immune response activation to increase the therapeutic effect of MSCs [136]. In another study, Wu H et al. [126] used EVs from miR-146a-genetically overexpressed MSCs. miR-146 acts as a negative feedback regulator of the innate immune response by targeting two adapter proteins, TNF receptor associated factor (TRAF) 6, and IL-1 receptor associated kinase (IRAK) [137]. Wu T et al. used MSCs derived from miR21-deficient mice, thus enhancing the expression of TGF-β1 that ultimately induced increased frequencies of regulatory T cells in vivo [138] (Table 7).

Other studies have pursued the overexpression of molecules that are responsible for in vivo trafficking of MSCs to the inflamed tissues. In this line, CXCR4 [133,139,140] in response to SDF-1 (CXCL12), or CX3CR1 in response to CX3CL1, have been tested in preclinical models of colitis [135]. Li X et al. [141] used intercellular adhesion molecule 1 (ICAM1)-overexpressed MSCs in a mouse model of colitis, aiming to guide the infused ICAM1-MSCs to inflamed intestinal tissue and to the spleen, thus improving their therapeutic effects. Alternatively, Yu T et al. [129] have used EVs from ephrin (Eph) B2-overexpressing MSCs as the released EphB2-carried EVs could migrate to distant tissues to exert their biological effects and decrease their retention within the spleen, lung, and liver after systemic administration (Table 7).

**Table 7 biomedicines-09-01507-t007:** Preclinical studies of IBD using genetically modified MSCs overexpressing different molecules (listed in alphabetical order).

Reference	Animal Model	Specie, Strain, Gender	MHC Context	Source of MSCs	Administration Route	MSC Dose	No. of MSCs Infusions	Genetically Modified MSCs	Day of Infusion (D)	Parameters Analysed for IBD Progression	Therapeutic Effect of MSC Therapy
Chen Y, 2015 [67]	7-day DSS cycle	Mouse, C57BL/6, F	X	UC	IV	1 × 10^6^	1	-WT MSCs -IFNγ-MSCs -IFNγ-pretreated MSCs	D8	-DAI (Body weight, stool consistency, and rectal bleeding) -Colon H/E staining -Th1, Th2, Th17, and Treg (mLN and SP)	Yes, IFNγ-MSCs > IFNγ-pretreated MSCs > WT MSCs
Chen Z, 2018 [140]	TNBS	Mouse, BALBC, F	S	BM	IV	1 × 10^6^	1	-WT MSCs -CXCR4-MSCs	24h	-DAI (Body weight, stool consistency, and rectal bleeding) -Colon H/E staining and IHC (occluding and VEGF)	Yes, CXCR4-MSCs = WT MSCs
Fu Y, 2020 [135]	7-day DSS cycle	Rat, Sprague Dawley, M	S	BM	IV	2.5 × 10^6^	3	-WT MSCs -CX3CR1-MSCs -IL-25-MSCs -CX3CR1 + IL-25-MSCs	D4, D6, and D8	DAI (Body weight and survival) -Colon length, H/E staining, and MPO activity -IFN-γ, IL-1β, IL-6, IL-12p70, IL-17A, IL-23, IL-25, RORγT T-bet, and TNF-α (colon) - IFN-γ and IL-17A in CD4 T cells (colon) -CD4, CD8, and CD19 (PB) - (IL-10, TNF-α, IFN-γ, IL-12p70, IL-6 and IL-1β (heart, lung, liver, SP, and kidney)	-Yes, CX3CR1 + IL-25-MSCs > IL25-MSCs = CXCR3-MSCs -No, WT MSCs
Kang J, 2019 [56]	10-day DSS cycle	Mouse, BALB/C, NA	X	UC	IP	3 × 10^6^	3	-WT MSCs -WT MSCs + miR148b-5p inhibitor -WT MSCs + 15-lox-1 -miR148b-5p-MSCs	D3, D6, and D9	-DAI (Body weight and stool consistency) -Colon length, H/E staining, and IHC (PCNA) -caspase 3, IL1β, IL6, 15lox1, PCNA, and TNFα (colon)	-Yes, miR148b-5p-MSCs = 15-lox-1-WT MSCs = WT MSCs -No, WT MSCs + miR148b-5p inhibitor
Kim DJ, 2012 [19]	8-day DSS cycle	Mouse, C57BL/6, NA	X	BM	SC	−1 × 10^5^ −3 × 10^4^	3	-IL-12p40-MSCs -WT MSCs -rAd/IL-12p40	-D0, D3, and D6 -D6, D9, and D12	-Body weight, stool consistency, and rectal bleeding -Colon length and H/E staining -IFNγ and IL17a in CD4 T cells (mLN and SP)	-Yes, IL12p40-MSCs = rAd/IL12p40 -Y, D0, D3, and D6 = D6, D9, and D12 -Yes, 1 × 10^5^ > 3 × 10^4^ -No, WT MSCs
Li X, 2019b [141]	7-day DSS cycle	Mouse, BALB/C, Mouse	S	BM	IV	1 × 10^6^	1	-WT MSCs -ICAM-1-MSCs	D3	-Body weight, stool consistency, recrtal bleeding, and survival -Colon H/E staining and IHC (TNFα) -IFN-γ, IL-4, and IL-17A in CD4 T and Treg cells (colon and SP)	Yes, ICAM-1-MSCs > WT MSCs
Liu L, 2012 [136]	8-day DSS cycle	Mouse, C57BL/6, NA	X	UC	IP	1 × 10^6^	2	-WT MSCs -pre-181a-MSC	D1 and D3	-Body weight, stool consistency, rectal bleeding, and survival -Colon length and H/E staining	-Yes, pre-181a-MSC -No, WT MSCs
Liu X, 2014 [139]	TNBS	Rat, Sprague Dawley, F	S	BM		2 × 10^6^	1	-WT MSCs -CXCR4-MSCs	D4	-DAI (Body weight, stool consistency, and rectal bleeding) -Colon length, H/E staining, and IHC (STAT3 and pSTAT3) -IFNγ, IL6, IL10, CXCR4, SDF-1a, STAT3, pSTAT3, and TNFα (colon)	-Yes, CXCR4-MSCs -No, WT MSCs
Nan Z, 2018 [133]	TNBS	Rat, Sprague Dawley, M	S	BM	IV	5 × 10^6^	1	-WT MSCs -CXCR4 + IL-35 MSCs	D3	-Body weight, stool consistency, and rectal bleeding -Colon length and H/E staining -FOXP3, IL-10, IL-17A, IL-35, and RORγT, (colon, mLN and SP) -Th17 and Treg (mLN and SP)	Yes, CXCR4 + IL35 > WT MSCs
Wang WQ, 2015 [132]	8-day DSS cycle	Mouse, C57BL6, F	S	BM	IP	0.5 × 10^6^	1	-WT-MSCs -IL37b-MSCs	D0	-Body weight -Colon length, and H/E staining -Gr1^+^CD11b^+^, Th1, Th2, and Treg (SP)	Yes, IL37b-MSCs > WT MSCs
Wu H, 2019 [126]	TNBS	Rat, Sprague Dawley, M	S	BM	IV	100 µg	1	-EVs from WT MSCs -EVs from miR-146a-MSCs	D3	-DAI (Body weight, stool consistency, and rectal bleeding) -Colon length and H/E staining -IκBα, IL-1β, IL-6, IL-10, IRAK1, NF-κBp65, TNF-α, and TRAF6 (colon)	Yes, EVs from miR-146a-MSCs >EVs from WT MSCs
Yan Y, 2018 [134]	7-day DSS cycle	Mouse, C57BL6, M	S	AD	IV	1 × 10^6^	3	-WT MSCs -IL-35-MSCs	D2, D4, and D6	-Body weight, stool consistency, and rectal bleeding -Colon length and H/E staining -IFNγ, IL17, TNF, and Treg (colon)	-Yes, IL35-MSCs -No, WT MSCs
Yu T, 2021 [129]	7-day DSS cycle	Rat, Sprague Dawley, M	S	BM	IV	100 µg	1	-EVs from WT MSCs -EVs from EPhB2-MSCs -Mesalazine	D8	-DAI -Colon H/E staining, length, and MPO activity -EphB2, eprhin-B1, FOXP3, GSH, IFNγ, IL1β, IL2, IL6, IL10, IL17, MDA, NFκB, pNFκB, occcludin, RORγT, SOD, pSTAT3, STAT3, TFG β, TNFα, and ZO1 (colon) -Th17 and Treg (SP)	Yes, EVs from EPhB2-MSCs = mesalazine > EVs from WT MSCs

Therapeutic effect; > better than, = similar to, < less than.

Strikingly, Chen Y et al. [67] used IFN-γ-plasmid transfected-MSCs and observed better therapeutic effects than native MSCs and IFN-γ-pretreated-MSCs. In this study, the authors demonstrated that IFN-γ enhances MSCs to express chemokine receptors, intercellular adhesions, and IDO, and also influences the differentiation of monocytes, activity of natural killer cells (NK) killing, and modulation of T cell response (Table 7).

All of these proposed genetically induced modifications of native MSCs have proven to efficiently increase the beneficial effects of MSCs in preclinical models of IBD, which suggests that the potential translation of these approaches to the clinic could be feasible. Until now, no clinical trials with genetically modified MSCs in any immune-mediated disease have been conducted or are registered.

## 5. Concluding Remarks and Future Perspectives

MSC-based therapy modulates intestinal inflammation when MSCs are infused at early phases of the disease, regardless of the tissue source, MHC context, or route of administration in most of the preclinical models of IBD. No adverse effects were observed in any case. Despite these successful preclinical results in IBD, no consensus on the efficacy of intravenous infusion of MSCs in IBD clinical trials can be yet drawn, mostly due to the heterogeneity of protocols, tissue sources, cell dosage, and disease activity status parameters used in addition to the limited number of IBD patients and control groups included. Furthermore, and in contrast to preclinical studies in IBD, the completed IBD clinical trials with MSC-based therapy have been carried out in refractory IBD patients, which may have hindered in part the beneficial effects of the MSC-based therapy. In ongoing clinical trials, IBD patients in early stages of the disease are being recruited, which may increase the efficacy of MSC-based therapy. There is an increasing number of preclinical studies in IBD which seek to further optimize MSC-based cell therapy, utilizing different culture media or pretreatments with small molecules and cytokines during ex vivo MSC expansion and combination with current IBD treatments and genetic engineering of MSCs. These alternative approaches have demonstrated enhanced efficacy of MSC-based cell therapy in preclinical studies of IBD, which shows the potential for these novel strategies to increase the therapeutic effects of MSC-based therapies. Further studies are currently being conducted to ensure the safe translation of these therapies to the clinic for the benefit of those IBD patients not responding to current treatments. Moreover, assessment of long-term efficacy of MSC-based therapies also needs to be addressed in the near future. All of these novel approaches and the successful results in preclinical and clinical studies will allow further development of mesenchymal stem/stromal cell-based therapy as a feasible alternative therapy for inflammatory bowel disease patients with unmet medical needs.

## Figures and Tables

**Table 3 biomedicines-09-01507-t003:** Preclinical studies of MSC-based therapy in IBD in combination with other cell-based therapies (listed in alphabetical order).

Reference	Animal Model	Specie, Strain, Gender	MHC Context	Source of MSCs	Administration Route	MSC Dose	No. of Infusions	Combined Cell Therapy Used	Day of Infusion (D)	Parameters Analysed for IBD Progression	Therapeutic Effect of MSC Therapy
Abbasi-Kenarsari H, 2020 [94]	7-day DSS cycle	Mouse, C57BL6/J, F	S	AD	IP	−1 × 10^6^ −5 × 10^5^	1	-WT MSCs -WT MSCs + Tol-DCs -Tol-DCs	D2	-DAI (Body weight, stool consistency, and rectal bleeding) -Colon length and weight, MPO activity, and H/E staining -Treg and cytokine levels (mLN and SP)	Yes, Tol-DCs + MSCs >TolDCs > WT MSCs
Liu X, 2015 [95]	TNBS	Mouse, BALBC, NA	S	BM	IP	3 × 10^6^	3	-WT MSCs + HSCs	D6, D4, and D0	-Colon H/E staining -IL10, IL12, and TNFα (serum) -Treg, CD4^+^ T, and IRF8^+^CD11b^+^ (SP)	Yes
Wei Y, 2009 [96]	TNBS	Rats, Sprague Dawley rats, M	S	BM	IV	2 × 10^6^	1	-WT MSCs -WT MSCs + HSCs -HSCs	24 h	-Colon histological analysis (BrDU and SRY)	Yes, WT MSCs + HSCs > MSCs = HSCs
Yu Y, 2017a [97]	10-day DSS cycle	Mouse, C57BL6, NA	S	BM	IV	1 × 10^6^	1	-WT MSCs -WT MSCs + Tregs	D3	-DAI (Body weight and survival) -Colon length and H/E staining -IFNγ, IL1, IL10, IL17a, and TNFα (colon) -Apoptotic CD3^+^ T cells and TGFβ (serum)	Yes, WT MSCs + Tregs > WT MSCs = Tregs

Therapeutic effect; > better than, = similar to, < less than.

## Data Availability

Not applicable.

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
