# Peer review of "Improving the Efficacy of Mesenchymal Stem/Stromal-Based Therapy for Treatment of Inflammatory Bowel Diseases"

_biomedicines, 2021, doi:10.3390/biomedicines9111507_

Round 1

Reviewer 1 Report

An interesting and very complete review about the use of Mesenchymal/Stromal-Based therapies in the management of inflammatory bowel diseases. I found the paper very informative and ideal for the one who approached for the first time the argument. I think only minor revisions will be necessary prior to the acceptance of this paper.

I think that a conclusion paragraph better focusing on the future perspective of the use of this kind of treatment in the management of IBD would be a great addition to the paper.

page 2 line 15-23 "Current treatments for IBD aim to decrease inflammation to prevent recurrence and to prolong periods of remission. These treatments include untargeted therapies such as aminosalicylates (principally mesalazine), 
glucocorticoids (conventional and other forms like budesonide or beclomethasone), antibiotics (typically ciprofloxacin and metronidazole) 
and immunosuppressants (mostly azathioprine/6-mercaptopurine or 
methotrexate) as well as targeted therapies such as biologic therapies 
which principally include TNFα inhibitors (infliximab, adalimumab, certolizumab pegol and golimumab) and blockers of signal transduction cascades (JAK inhibitors)." these paragraph needs some reference, such as: doi: 10.1080/03007995.2020.1786681. and doi: 10.1111/dth.12811. 

Thank You

Author Response

An interesting and very complete review about the use of Mesenchymal/Stromal-Based therapies in the management of inflammatory bowel diseases. I found the paper very informative and ideal for the one who approached for the first time the argument. I think only minor revisions will be necessary prior to the acceptance of this paper.

I think that a conclusion paragraph better focusing on the future perspective of the use of this kind of treatment in the management of IBD would be a great addition to the paper.

We appreciate the Reviewer´s comments on this. In fact, the ‘Discussion’ of the submitted manuscript is actually dedicated to draw conclusions based on preclinical and clinical data and on future prospects of MSC-based therapies. In the revised manuscript, the Discussion has been renamed as ‘Concluding remarks and future perspectives’. In addition to this, the following paragraph has been included: ‘All of these novel approaches and the successful results in preclinical and clinical studies will allow further development of mesenchymal stem/stromal cell-based therapy as a feasible alternative therapy for inflammatory bowel disease patients with unmet medical needs’.

page 2 line 15-23 "Current treatments for IBD aim to decrease inflammation to prevent recurrence and to prolong periods of remission. These treatments include untargeted therapies such as aminosalicylates (principally mesalazine), glucocorticoids (conventional and other forms like budesonide or beclomethasone), antibiotics (typically ciprofloxacin and metronidazole) and immunosuppressants (mostly azathioprine/6-mercaptopurine or methotrexate) as well as targeted therapies such as biologic therapies which principally include TNFα inhibitors (infliximab, adalimumab, certolizumab pegol and golimumab) and blockers of signal transduction cascades (JAK inhibitors)." these paragraph needs some reference, such as: doi: 10.1080/03007995.2020.1786681. and doi: 10.1111/dth.12811. 

We very much appreciate the Reviewer´s suggestion and accordingly, Roberti R et al´s and Spagnuolo R et al´s papers have been included in Introduction (reference numbers 4 and 5, respectively) in the revised manuscript.

Reviewer 2 Report

This is a comprehensive and well-organized review of the previous studies for mesenchymal stem cell (MSC)-based therapy for treatment of inflammatory bowel diseases (IBDs) through an extensive literature review.

  1. TALBE is too broad, so I hope it will be made simple. Parameters analyzed IBD progression is important, but it would be nice to exclude it because its importance decreases relatively.
  1. This review article focused on preclinical studies; however, I think that the results of the human clinical trials should be summarized due to practical and clinical application. Actually, the authors summarized human clinical trials in Supplementary Table 3. Readers are also likely to expect the section of the review for the results of the human clinical trials.

Author Response

This is a comprehensive and well-organized review of the previous studies for mesenchymal stem cell (MSC)-based therapy for treatment of inflammatory bowel diseases (IBDs) through an extensive literature review.

  1. TABLE is too broad, so I hope it will be made simple. Parameters analyzed IBD progression is important, but it would be nice to exclude it because its importance decreases relatively.

We very much appreciate Reviewer 2’s general comments on the manuscript. We are uncertain on whether we understood correctly Reviewer’s comments as it is unclear to us the what table Reviewer 2 is referring to. We honestly think that all the parameters included in all tables will be very useful for the readers aiming to have a glimpse on mesenchymal stem cell (MSC)-based therapy for treatment of inflammatory bowel diseases (IBDs). We are opened to discuss this further if necessary.

This review article focused on preclinical studies; however, I think that the results of the human clinical trials should be summarized due to practical and clinical application. Actually, the authors summarized human clinical trials in Supplementary Table 3. Readers are also likely to expect the section of the review for the results of the human clinical trials.

We very much appreciate Reviewer’s comment on this. As suggested, we could move Supplementary Table 3 to the main text although this table would be the first one to be mentioned as ‘Table 1’ that, in fact, is formed by Table1a and Table1b in the revised manuscript. A detailed review of clinical studies is out of the scope of the present work so we would rather prefer to maintain this table in Supplementary information. In any case, we are opened to Editor’s and Reviewer’s choice.